# Bacterial Pathogens and Evaluation of a Cut-Off for Defining Early and Late Neonatal Infection

**DOI:** 10.3390/antibiotics10030278

**Published:** 2021-03-09

**Authors:** Pavla Kucova, Lumir Kantor, Katerina Fiserova, Jakub Lasak, Magdalena Röderova, Milan Kolar

**Affiliations:** 1Department of Microbiology, Faculty of Medicine and Dentistry, Palacky University Olomouc and University Hospital Olomouc, 779 00 Olomouc, Czech Republic; pavla.kucova@fnol.cz (P.K.); katerina.fiserova@fnol.cz (K.F.); magdalena.roderova@upol.cz (M.R.); milan.kolar@fnol.cz (M.K.); 2Neonatal Department, University Hospital Olomouc, 779 00 Olomouc, Czech Republic; jakub.lasak@fnol.cz

**Keywords:** newborn, infection, bacteria, antibiotic therapy

## Abstract

Bacterial infections are an important cause of mortality and morbidity in newborns. The main risk factors include low birth weight and prematurity. The study identified the most common bacterial pathogens causing neonatal infections including their resistance to antibiotics in the Neonatal Department of the University Hospital Olomouc. Additionally, the cut-off for distinguishing early- from late-onset neonatal infections was assessed. The results of this study show that a cut-off value of 72 h after birth is more suitable. Only in case of early-onset infections arising within 72 h of birth, initial antibiotic therapy based on gentamicin with ampicillin or amoxicillin/clavulanic acid may be recommended. It has been established that with the 72-h cut-off, late-onset infections caused by bacteria more resistant to antibiotics may be detected more frequently, a finding that is absolutely crucial for antibiotic treatment strategy.

## 1. Introduction

Neonatal infections may be defined as infectious diseases occurring in newborns within 4 weeks after birth. They may be classified as early-onset or late-onset, with the cut-off for distinguishing early- from late-onset infections ranging between 72 h and 7 days.

Early-onset neonatal infections are caused by microorganisms transmitted in utero or as the baby moves down the birth passage (antepartum or intrapartum), with 85% of them occurring within 24 h from birth [1,2]. The most common etiologic agents are *Escherichia coli*, *Streptococcus agalactiae* (GBS), and *Listeria monocytogenes*. Less frequently, chlamydias and mycoplasmas are shown to play an etiologic role [3,4,5]. For initial antibiotic therapy, ampicillin or amoxicillin with clavulanic acid, combined with gentamicin are recommended [2,6,7]. In the literature, however, information is available about increasing resistance of *Escherichia coli* to antibiotics including aminopenicillins combined with inhibitors of bacterial beta-lactamases (ampicillin/sulbactam and amoxicillin/clavulanic acid) and potential failure of antibiotic therapy [8,9,10,11]. The risk factors for the development of early-onset infections include vaginal colonization with GBS, prelabor rupture of membranes, prematurity, multiple abortions, maternal malnutrition, and congenital abnormalities [12,13,14].

Late-onset neonatal infections are caused by bacteria associated with hospital care. Multidrug-resistant bacteria are frequently responsible for these infections. Important sources are artificial materials, in particular, cannulas or catheters [13,15]. The most common bacterial pathogens are *Staphylococcus aureus*, coagulase-negative staphylococci, *Klebsiella pneumoniae*, *Escherichia coli, Enterobacter* spp., *Serratia marcescens*, *Pseudomonas aeruginosa*, *Acinetobacter baumannii*, anaerobic bacteria, and yeasts [16,17,18]. The risk factors for acquiring infections include prematurity, congenital abnormalities, and invasive care such as central venous catheter placement or respiratory support [13,19]. In many cases, antibiotic therapy requires application of antibacterial drugs effective against multidrug-resistant bacteria such as carbapenems, glycopeptides, and aminoglycosides in relevant combinations [20].

A very important part of the overall therapeutic approach to neonatal infections is microbiological examination of adequate clinical materials, in particular, blood and cerebrospinal fluid cultures. The obtained results allow targeted antibiotic therapy based on identification of bacterial pathogens and determination of their susceptibility/resistance to antibiotics. Under defined conditions, nasopharyngeal and rectal swabs, pharyngeal aspirate, urine, and other specimens may also be of clinical importance if very carefully interpreted [21].

Our study aimed to assess the cut-offs for distinguishing early- from late-onset neonatal infections and determine the most frequent bacterial pathogens causing or suspected of causing neonatal infections, including their resistance to antibacterial drugs.

## 2. Results

Over a 3-year period, a total of 7221 newborns were either born or transported to the Neonatal Department of the University Hospital Olomouc. Neonatal infections were treated in 364 cases (5%), in whom 263 bacterial and 59 mycotic pathogens were isolated.

Clinical sepsis (72%) was most common in newborns, followed by bloodstream infections (13%) and pneumonia (9%); less frequent were meningitis (1%) and necrotizing enterocolitis (1%). In 4% of newborns treated for infections of unclear origin, no bacterial pathogens were isolated.

The sample consisted of the following gestational age subgroups: term 38%, moderate to late preterm 28%, very preterm 20% and extremely preterm newborns 14%. The birth weight subgroups were as follows: normal birth weight 41%, low birth weight 26%, very low birth weight 14% and extremely low birth weight 19%.

The most frequently isolated bacteria were *Escherichia coli* (22%), *Klebsiella pneumoniae* (12%), coagulase-negative staphylococci (12%), and *Staphylococcus aureus* (10%) (Table 1). The results show that *Enterobacterales* accounted for more than half (56%) of all bacterial pathogens. *Streptococcus agalactiae* caused infection in only nine newborns and *Listeria monocytogenes* was detected in a single infant. Besides bacterial pathogens, *Candida* spp. were responsible for 59 cases of infections (majority of which was treated topically), with *Candida albicans* causing infections in 47 (<1%) newborns. *Candida parapsilosis* and *Candida tropicalis* were confirmed as etiologic agents in only 7 and 5 neonatal infections, respectively.

Overall, 6% of newborns in 2015 and 5% in both 2016 and 2017 were treated with antibiotics for confirmed or suspected early- or late-onset infection. Table 2 shows bacterial pathogens causing early- and late-onset infections as defined by the two cut-offs (72 h and 7 days).

When the 72-h cut-off for developing symptoms of early-onset infection was applied, a total of 57 pathogens were isolated, mostly enterobacteria (58%). The most frequently isolated species were *Escherichia coli* (26%) and *Klebsiella pneumoniae* (18%). The use of the other cut-off, i.e., early-onset infections in the first 7 days of life, resulted in isolation of 139 pathogenic bacteria. The most common organisms were enterobacteria (47%), mainly *Escherichia coli* (19%) and *Klebsiella pneumoniae* (14%), followed by Gram-positive bacteria (31%), mainly coagulase-negative staphylococci (11%) and *Staphylococcus aureus* (9%).

In case of late-onset infections defined by the 72-h cut-off, a total of 206 bacterial pathogens were isolated. The most common were enterobacteria (48%), in particular, *Escherichia coli* (20%), followed by *Klebsiella pneumoniae* (10%) and *Enterobacter cloacae* (5%). Further, there were 28% of Gram-positive bacteria (coagulase-negative staphylococci in 13% and *Staphylococcus aureus* in 11%). When late-onset neonatal infections were defined by symptoms occurring from the eighth day, 124 bacterial pathogens were identified; the most frequent species were *Escherichia coli* (25%) and *Klebsiella pneumoniae* (10%).

Assessment of the two cut-offs (i.e., ≤72 h and ≤7 days of life) showed that the presence changed for most bacterial species. This was obvious in case of coagulase-negative staphylococci; when the 72-h cut-off was applied, only 4 out of 31 cases could be characterized as early-onset, as compared with 15 cases when using the 7-day cut-off. Similar differences were observed for *Staphylococcus aureus*, *Escherichia coli,* and *Klebsiella pneumoniae* (Figure 1). Yet another example may be non-fermenting Gram-negative bacteria. Out of 14 cases with *Pseudomonas aeruginosa* infection, 1 and 8 were interpreted as early-onset according to the 72-h and 7-day cut-offs, respectively. Similarly, among 15 cases in which *Stenotrophomonas maltophilia* was isolated, only 1 was early-onset when the 72-h cut-off was used, as compared with 7 cases when applying the other cut-off. In case of *Candida albicans* infections, it is clear that according to the 72-h cut-off, only 3 out of 47 cases could be defined as early-onset, while with the 7-day cut-off, it was 24 cases (Figure 1).

Table 3 shows the numbers of suspected or confirmed causative bacteria isolated from newborns treated with antibiotics as well as the numbers of newborns in whom no bacterial pathogen was detected. According to the results, only one bacterial species was isolated in 76% of newborns, more than one microorganisms were isolated in 9% and no etiologic agent was identified in 15% of infants with clinical sepsis.

The results suggest relatively good susceptibility of the most frequently isolated enterobacteria (*Escherichia coli*, *Klebsiella pneumoniae*, *Enterobacter cloacae,* and *Klebsiella oxytoca*) to antibacterial drugs with the exception of ampicillin and cefazolin. Production of broad-spectrum beta-lactamases (only ESBL and AmpC types) was confirmed in eight strains, i.e., 6% of isolated enterobacteria (Table 4). All those strains were isolated from neonates who developed infection from day 4 of life onwards. Resistance to meropenem was not noted.

Pulsed-field gel electrophoresis (PFGE) of 3 CTX-M-15-positive *Klebsiella pneumonia* and 2 CTX-M-15-positive *Escherichia coli* strains showed that those were strains with unique restriction genetic profiles. Therefore, clonal horizontal transmission was ruled out (Figure 2 and Figure 3).

The most frequently isolated Gram-positive bacteria, *Staphylococcus aureus*, was 100% susceptible to oxacillin. It means that methicillin-resistant *Staphylococcus aureus* (MRSA) was not detected. Very high susceptibility was found to ciprofloxacin, erythromycin, clindamycin, gentamicin, and glycopeptides (vancomycin and teicoplanin). Higher resistance levels were documented in coagulase-negative staphylococci, but vancomycin and teicoplanin were found to be very effective. Isolated strains of *Enterococcus faecalis* and *Streptococcus agalactiae* were 100% susceptible to ampicillin and glycopeptides (vancomycin and teicoplanin).

Susceptibility of the most frequently isolated bacterial pathogens to antibiotics is illustrated in Table 5 and Table 6.

## 3. Discussion

An indispensable part of the therapeutic approach to neonatal infections is application of antibacterial drugs immediately after establishing the diagnosis or as soon as strong suspicion arises. Antibiotic regimens should consider microbiological examinations of the newborn and its mother as well as local results of surveillance of the most common bacterial pathogens and their resistance to antibiotics. Based on microbiology test results, an integral part of the overall therapeutic approach, targeted antibiotic therapy should be administered.

Over the study period, a total of 7221 newborns were hospitalized in the Neonatal Department of the University Hospital Olomouc, with bacterial infections being treated in 5% of them. This percentage is very high for several reasons. Some of the neonates were transported to our tertiary neonatal unit due to serious health complications. As some preterm neonates were treated more than once, the percentage reflects the number of treated cases rather than the number of treated children. Therefore, the rate is not the incidence of neonatal infections, but the incidence of antibiotic treatment, preemptive in some cases. The study results suggest that the most common bacterial agents causing or suspected of causing neonatal infections were enterobacteria (52% of all isolated pathogens) and staphylococci (21%). The most frequent causative species were *Escherichia coli* (22%)*, Klebsiella pneumoniae* (12%), coagulase-negative staphylococci (12%), and *Staphylococcus aureus* (10%). *Candida* spp. infections were caused by *Candida albicans* in the vast majority of cases (80%). These results are consistent with those published by Russell who found that in neonatal infections, the most common isolates were *Escherichia coli*, coagulase-negative staphylococci, *Staphylococcus aureus,* and *Enterococcus* spp. [22]. Shane et al. reported that in neonatal infections, *Streptococcus agalactiae* (43%) and *Escherichia coli* (29%) were most frequently isolated. These species belong to the most important pathogens contributing to the development of early-onset neonatal infections. According to the authors, late-onset infections were caused by *Staphylococcus aureus,* coagulase-negative staphylococci, enterococci, enterobacteria, and *Streptococcus pyogenes* [23]. In a study by Bulkowstein et al., the most common pathogens causing early-onset infections were *Escherichia coli* (34%), *Streptococcus agalactiae* (22%), *Klebsiella pneumoniae* (13%), *Enterococcus faecalis* (9%), and *Staphylococcus aureus* (5%). The spectrum of pathogens was similar for late-onset infections; most frequently, the authors isolated *Escherichia coli* (29%), *Streptococcus pneumoniae* (13%), *Staphylococcus aureus* (9%), and *Klebsiella pneumoniae* (7%) [24].

Given the present results, the 72-h cut-off appears to be more suitable for distinguishing early- from late-onset neonatal infections. Therefore, the use of a different antibiotics could be advised. Within 72 h of birth, microorganisms transmitted from the mother in utero or during birth were isolated more frequently. However, when infections developing in the first 7 days of life were considered as early-onset, more bacteria associated with nosocomial infections were isolated. Our results indicate that with the 72-h cut-off, bacteria originating from the mother (mainly *Streptococcus agalactiae*) were detected. At the same time, the frequency of non-fermenting bacteria (e.g., *Stenotrophomonas maltophilia* and *Pseudomonas aeruginosa*) was very low. With the 7-day cut-off for early neonatal infections, almost a double number of bacterial pathogens was identified. These bacteria show higher levels of resistance to antibiotics, and they are very often associated with hospital care and nosocomial infections. It should be emphasized that infections caused by enterobacteria producing broad-spectrum beta-lactamases would also fall into the category of early-onset infections if the 7-day cut-off was used. However, if the 72-h cut-off was applied, all these infections would be characterized as late-onset. Alternatively, bacterial strains were detected that tend to be related to the use of long-term central venous catheters, in particular coagulase-negative staphylococci. It has been established that with the 72-h cut-off, late-onset infections caused by bacteria more resistant to antibiotics may be detected more frequently, a finding that is absolutely crucial for antibiotic treatment strategy.

Infections caused by *Candida* spp. are associated with significant morbidity and mortality in infants. As with bacterial infections, premature babies with extremely low birth weight (less than 1000 g) are most at risk of mycotic infections [25]. A rather serious disease of premature neonates is candidiasis, which can manifest as candidemia, urinary tract infection, or as impairment of any tissue or structure. The risk factors for developing candidiasis include prematurity, central vascular catheterization, abdominal surgery, necrotizing enterocolitis, exposure to broad-spectrum antibacterial agents (e.g., third-generation cephalosporins and carbapenems), parenteral nutrition, antacid use, and endotracheal intubation [26]. The use of catheters or endotracheal tubes destroys the body’s natural barriers, allowing yeasts to penetrate, multiply, and invade sterile body areas. It is, therefore, clear that mycotic infection would not be possible without prior colonization. According to Roilides, the species most commonly occurring in pediatric patients are *Candida albicans* (50%), *Candida parapsilosis* (21%), and *Candida tropicalis* (10%) [27]. The same species were also most frequent in the present study. The highest rates of yeasts were noted after the first 72 h of life. In these cases, it is more preferable to use the 72-h cut-off for early neonatal infections, since almost 10 times less yeasts were detected compared to the 7-day cut-off.

In the present study, GBS accounted for a very small proportion (3%) of bacterial pathogens identified over the 3-year period. This species mainly plays a role in early-onset infections. This fact was confirmed by the present study, with 8 out of 9 GBS strains being isolated in the first 72 h of life. It may be calculated that GBS neonatal infection affected 1.2 newborns per 1000 live births. According to Strakova et al., the incidence of invasive streptococcal neonatal infections ranged from 0.7 to 1.0 cases per 1000 live births in 2001 and 2002 [28]. Simetka et al. reported early-onset GBS infections in 1.2 newborns per 1000 live births in 2003–2005 [29]. These findings may be explained by very effective prophylaxis as early as in the prenatal period, that is screening of all pregnant women between 35 and 37 weeks of gestation, early laboratory diagnosis and relevant prepartum/intrapartum antibiotic prophylaxis [30,31,32].

In the present study, antimicrobial susceptibility testing showed relatively high efficacy of antibiotics. No MRSA strains or vancomycin-resistant enterococci were detected. Production of broad-spectrum beta-lactamases (CTX-M-15, CTX-M-9, EBC, and CIT) was confirmed in only 6% of enterobacteria with unique genetic profiles; no clonal spread was confirmed. On the other hand, a considerable therapeutic problem is posed by multidrug-resistant strains of *Stenotrophomonas maltophilia* and *Burkholderia cepacia* complex susceptible to antibiotics contraindicated in neonatal care.

The main limitation of the study is its retrospective character. In addition, data from only one neonatal center were processed. It is most crucial to follow local antimicrobial resistance surveillance results when choosing initial antibiotic treatment.

## 4. Material and Methods

The primary outcome of the study is to evaluate the cut-offs for distinguishing early- from late-onset neonatal infections. In neonatology, there are two most used cut-offs. They are 72 h of life and 7 days of life [1,2,6,11,13]. The aim of the study was to evaluate these existing and widely used cut-offs, based on bacteria isolated in specific postnatal age. We did not want to suggest a new cut-off. The secondary outcome can be characterized as the determination of the most frequent bacterial/mycotic pathogens causing or suspected of causing neonatal infections, including resistance to antibacterial drugs.

Biological samples (urine, stools, blood cultures, venous cannulas, cerebrospinal fluid, bronchoalveolar lavage fluid, axillary, throat, nasal, ear, conjunctival, and wound swabs) were obtained from newborns before antibiotic treatment. All biological samples were collected only as part of standard clinical care and active bacterial surveillance program.

Patients’ parents gave informed consent to hospitalization, sample collection, and anonymous enrollment in the study. Ethics committee approval was not required as the study did not interfere with the diagnostic and therapeutic process.

The case inclusion criterion was antibiotic treatment, irrespective of infection type, provided to newborns staying in the Neonatal Department of the University Hospital Olomouc in 2015–2017. No patient treated with antibiotics was excluded from the study. To define infections, the modified NEO-KISS criteria were used for early- as well as late-onset sepsis.

Bacterial isolates were identified by MALDI-TOF MS (Biotyper Microflex, Bruker Daltonics, Bremen, Germany) [33]. Each identified strain was included only once in the database.

Bacteria isolated from blood or cerebrospinal fluid were considered to be etiologic agents. If a newborn was treated with antibiotics and bacteria were isolated from other samples, the bacteria were identified as a suspected etiologic agent.

Susceptibility to antibiotics was determined by using the microdilution method in accordance with the EUCAST recommendations [34]. Production of ESBLs and AmpC-type beta-lactamases was detected by relevant phenotypic tests and confirmed by PCR detecting genes specific for particular beta-lactamase types [35].

The similarity of 3 CTX-M-15-positive *Klebsiella pneumoniae* and 2 CTX-M-15-positive *Escherichia coli* isolates was assessed with PFGE. Bacterial DNA was isolated using a technique described by Husickova et al. and digested by the *Xba*I restriction endonuclease (New England Biolabs, Ipswich, MA, USA) for 24 h at 37 °C [36]. The obtained DNA fragments were separated by PFGE on 1.2% agarose gel for 24 h at 6 V/cm and pulse times of 2–35 s. Subsequently, the gel was stained with ethidium bromide. The resulting restriction profiles were analyzed with the GelCompar II software (Applied Maths, Kortrijk, Belgium) using the Dice coefficient (1.2%) for comparing similarity and unweighted pair group method with arithmetic means for cluster analysis. The results were interpreted according to criteria described by Tenover et al. [37].

Statistical analyses were conducted using IBM SPSS Statistics version 22 (IBM, New York City, NY, USA) and the statistical environment R, version 4.0.2. The presence of bacteria assessed according to both cut-offs was compared with the McNemar’s one-sided test. The tests were performed at a 0.05 level of significance.

## 5. Conclusions

The results clearly show that the therapeutic approach to neonatal infections must be based on their character including classification as early-/late-onset. As a cut-off value, 72 h after birth are more suitable. At the same time, it must be stressed that microbiological tests are necessary to allow targeted antibiotic therapy including application of adequate antibiotics if multidrug-resistant bacterial pathogens are detected. The results of this study confirm that only in case of early-onset infections arising within 72 h of birth, initial antibiotic therapy based on gentamicin with ampicillin or amoxicillin/clavulanic acid may be recommended. It is also clear that initial antibiotic treatment must be based on local surveillance of the most common bacterial pathogens and their resistance to antibiotics.

## Figures and Tables

**Figure 1 antibiotics-10-00278-f001:**
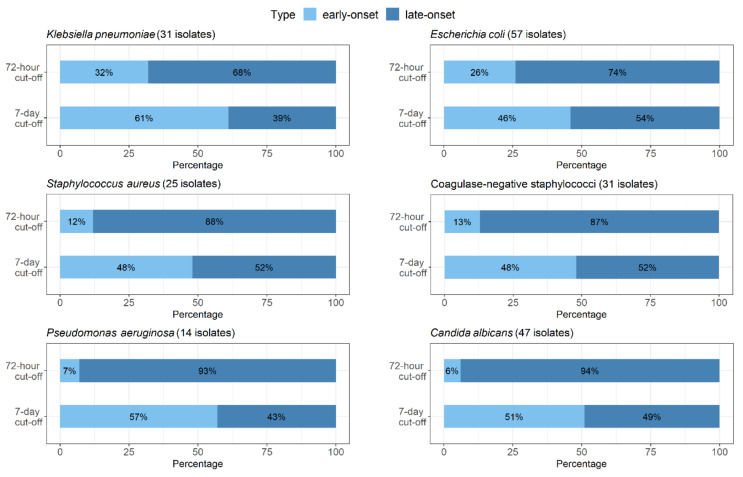
Proportional representation of early- and late-onset infections with regard to both cut-offs (72-h and 7-day) shown for *Klebsiella pneumoniae*, *Escherichia coli*, *Staphylococcus aureus*, coagulase-negative staphylococci, *Pseudomonas aeruginosa,* and *Candida albicans*.

**Figure 2 antibiotics-10-00278-f002:**
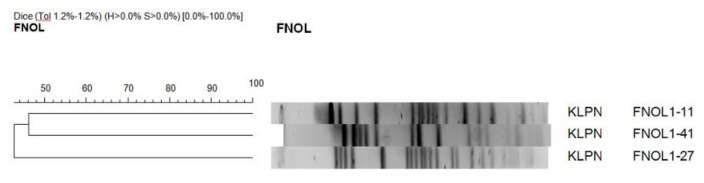
Pulsed-field gel electrophoresis (PFGE) of 3 CTX-M-15-positive *Klebsiella pneumonia* strains.

**Figure 3 antibiotics-10-00278-f003:**
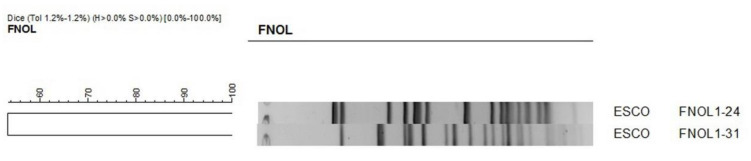
PFGE of 2 CTX-M-15-positive *Escherichia coli* strains.

**Table 1 antibiotics-10-00278-t001:** Bacteria isolated from newborns prior to initiation of antibiotic therapy.

Bacterial Species	No. of Isolates	Percentage
*Escherichia coli*	57	22
*Klebsiella pneumoniae*	31	12
Coagulase-negative staphylococci	31	12
*Staphylococcus aureus*	25	10
*Enterobacter cloacae*	16	6
*Klebsiella oxytoca*	16	6
*Stenotrophomonas maltophilia*	15	6
*Pseudomonas aeruginosa*	14	5
*Enterococcus faecalis*	11	4
*Streptococcus agalactiae*	9	3
*Burkholderia cepacia* complex	8	3
*Citrobacter freundii*	4	2
*Ralstonia picketii/insidiosa*	4	2
*Pseudomonas putida*	2	1
*Acinetobacter baumannii*	2	1
*Klebsiella aerogenes*	2	1
*Enterobacter kobei*	2	1
*Serratia marcescens*	2	1
*Haemophilus influenzae* type b	2	1
*Enterobacter asburiae*	2	1
Others	8	3

Legend: others—bacteria isolated only once (*Salmonella* Enteritidis, *Providencia rettgeri*, *Proteus mirabilis*, *Morganella morganii*, *Listeria monocytogenes*, *Streptococcus anginosus*, *Streptococcus intermedius*, and *Enterococcus faecium*).

**Table 2 antibiotics-10-00278-t002:** Numbers of bacterial/mycotic pathogens from newborns treated with antibiotics with regard to different cut-offs for distinguishing between early- and late-onset infections.

Bacterial Pathogen	Early-Onset (Absolute No.)		Late-Onset (Absolute No.)		
	≤72 h	≤7 days	>72 h	>7 days	*p*-Value
*Escherichia coli*	15	26	42	31	**<0.001**
*Klebsiella pneumoniae*	10	19	21	12	**0.002**
Coagulase-negative staphylococci	4	15	27	16	**<0.001**
*Staphylococcus aureus*	3	12	22	13	**0.002**
*Enterobacter cloacae*	5	9	11	7	0.063
*Klebsiella oxytoca*	2	7	14	9	**0.031**
*Stenotrophomonas maltophilia*	1	7	14	8	**0.016**
*Pseudomonas aeruginosa*	1	8	13	6	**0.008**
*Enterococcus faecalis*	3	7	8	4	0.063
*Streptococcus agalactiae*	8	9	1	0	0.500
*Burkholderia cepacia* complex	0	3	8	5	0.125
*Citrobacter freundii*	1	3	3	1	
*Ralstonia picketii/insidiosa*	0	3	4	1	
*Pseudomonas putida*	0	0	2	2	
*Acinetobacter baumannii*	0	1	2	1	
*Klebsiella aerogenes*	0	1	2	1	
*Enterobacter kobei*	0	0	2	2	
*Serratia marcescens*	0	0	2	2	
*Haemophilus influenzae* type b	1	2	1	0	
*Enterobacter asburiae*	0	1	2	1	
Other bacteria	3	6	5	2	
**Mycotic pathogen**	**≤72 h**	**≤7 days**	**>72 h**	**>7 days**	
*Candida albicans*	3	24	44	23	**<0.001**
*Candida parapsilosis*	0	3	7	4	0.125
*Candida tropicalis*	0	2	5	3	

Legend: *p*-values are given for bacterial and *Candida* species with a number of isolates > 5.

**Table 3 antibiotics-10-00278-t003:** Identification of etiologic agents causing/suspected of causing neonatal infections.

Etiology of Confirmed or Suspected Infections	2015 (Absolute No.)	2016 (Absolute No.)	2017 (Absolute No.)
No identified pathogen	19	23	14
Monomicrobial etiology	101	81	94
Polymicrobial etiology	12	9	11
Total	132	113	119

**Table 4 antibiotics-10-00278-t004:** Broad-spectrum beta-lactamase (ESBL- and AmpC only)-positive enterobacteria.

Species	No. of Strains	Type of Broad-Spectrum Beta-Lactamase
*Klebsiella pneumoniae*	3	ESBL—CTX-M-15
*Klebsiella pneumoniae*	1	ESBL—CTX-M-9
*Escherichia coli*	2	ESBL—CTX-M-15
*Enterobacter cloacae*	1	AmpC—EBC
*Citrobacter freundii*	1	AmpC—CIT

**Table 5 antibiotics-10-00278-t005:** Susceptibility of enterobacteria to selected antibiotics (percentages).

Antibiotic/Pathogen	*Escherichiacoli*	*Klebsiella pneumoniae*	*Klebsiella oxytoca*	*Enterobacter cloacae*
**AMI**	95	100	100	100
**AMP**	28	0	0	0
**AMS**	84	90	100	0
**CIP**	98	90	100	100
**COL**	100	87	94	81
**CPM**	98	90	100	94
**CRX**	95	90	100	0
**CTX**	95	90	100	81
**CTZ**	98	90	100	81
**CZL**	77	77	44	0
**GEN**	88	90	100	100
**MER**	100	100	100	100
**PPT**	96	90	100	81
**TOB**	88	90	100	100

Legend: AMI—amikacin, AMP—ampicillin, AMS—ampicillin/sulbactam, CIP—ciprofloxacin, COL—colistin, CPM—cefepime, CRX—cefuroxime, CTX—cefotaxime, CTZ—ceftazidime, CZL—cefazolin, GEN—gentamicin, MER—meropenem, PPT—piperacillin/tazobactam, TOB—tobramycin.

**Table 6 antibiotics-10-00278-t006:** Susceptibility of *Staphylococcus aureus* and coagulase-negative staphylococci to selected antibiotics (percentages).

Antibiotic/Pathogen	*Staphylococcus aureus*	Coagulase-Negative Staphylococci
**OXA**	100	28
**CIP**	100	57
**CLI**	80	51
**TEI**	100	93
**VAN**	100	93
**ERY**	80	30
**GEN**	80	45

Legend: OXA—oxacillin, CLI—clindamycin, TEI—teicoplanin, VAN—vancomycin, ERY—erythromycin.

## Data Availability

The support data are not publicly available due to privacy issues.

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
