# Peer review of "Bacterial Pathogens and Evaluation of a Cut-Off for Defining Early and Late Neonatal Infection"

_antibiotics, 2021, doi:10.3390/antibiotics10030278_

Round 1
Reviewer 1 Report
Dear Author
Thank you for your replay
I have only few comments :
A) Tab.1 line 85 Salmonella enteritidis instead of Enteritidis
B) Tab.2 Mycotic pathogen instead of fungal pathogen
C) line 234 Mycotic infection instead of fungal infection
Author Response
Reply to review report
The authors thank the reviewer for valuable comments. The manuscript has been modified based on them with one exception.
Comment: Tab.1 line 85 Salmonella enteritidis instead of Enteritidis.
Reply: The term Salmonella Enteritidis is really correct. The full name is Salmonella enterica subsp. enterica serotype Enteritidis and the abbreviated name Salmonella Enteritidis is commonly used. In this case, it is the designation of the serovar.
Comment: Tab.2 Mycotic pathogen instead of fungal patogen.
Reply: The manuscript has been modified in Table 2.
Comment: line 234 Mycotic infection instead of fungal Infection.
Reply: The manuscript has been modified in whole text (the term fungal has been replaced by the term mycotic).

Reviewer 2 Report
The authors investigated the cut-offs for distinguishing early- from late-onset neonatal infections and determined the most frequent bacterial pathogens causing neonatal infections, including their resistance to antibacterial drugs. They concluded that the therapeutic approach to neonatal infections must be based on their character including classification as early-/late-onset. Also, the results of the study confirmed that only in case of early-onset infections arising within 72 hours of birth, initial antibiotic therapy based on gentamicin with ampicillin or amoxicillin/clavulanic acid may be recommended.
The study is interesting, well designed and written. Still, there are several issues that need to be corrected and clarified before any favorable decision should be made.
My concerns are as follows:
- Primary and secondary outcomes of the study should be mentioned in the methodology.
- Inclusion / Exclusion criteria should be presented in methodology as well.
- Regardless of whether it is a retrospective study, the study should be approved by the ethics committee. Please provide IRB reference.
- How the parents signed informed consent if the study was retrospective?!
- Table 2 should be moved to page 3, and text between the Tables to page 2.
- It is unclear how the cut-offs were calculated. Which test was used for calculating the cut-offs? Was the ROC analysis performed? Please clarify
- Limitations of the study should be clearly stated at the end of discussion.
Author Response
Reply to review report
The authors thank the reviewer for valuable comments. The manuscript has been modified based on them.
Comment: Primary and secondary outcomes of the study should be mentioned in the methodology.
Reply: The manuscript has been modified according to the comment (lines 285-288).
Comment: Inclusion / Exclusion criteria should be presented in methodology as well.
Reply: Inclusion criteria are presented in paragraph ranging from line 297 to 301. As for exclusion criteria, no patient treated with antibiotics was excluded from the study. We completed the paragraph according to the comment.
Comment: Regardless of whether it is a retrospective study, the study should be approved by the ethics committee. Please provide IRB reference.
Reply: All bacterial pathogens were isolated from clinical samples obtained from patients solely as part of standard hospital care and their analysis was performed retrospectively using hospital information systems. In such cases, approvement by the ethics committee is not needed. However, in response to the reviewer’s comment, ethics committee was informed and we received a reply that approval was not required in this case. Similar examples are articles with the theme of resistance of bacterial pathogens to antibiotics if these were isolated within a standard hospital care.
Comment: How the parents signed informed consent if the study was retrospective?!
Reply: At admission to our Neonatal department, all parents are asked to sign written consent. Only if they agree, of course. The consent has been given by all participants´ parents.
It says “I do agree with storing and using biological samples, taken routinely as part of common diagnostic and therapeutic approaches, and using them for scientific purposes. I do agree with publication of obtained results, if anonymous.”
Comment: Table 2 should be moved to page 3, and text between the Tables to page 2.
Reply: It has been done.
Comment: It is unclear how the cut-offs were calculated. Which test was used for calculating the cut-offs? Was the ROC analysis performed? Please clarify
Reply: The cut-offs were not calculated. The two evaluated cut-offs are most commonly used in neonatology worldwide and there is no consensus, please see for example source:
Clinical features, evaluation, and diagnosis of sepsis in term and late preterm infants [online]. Uptodate, 2019 [cit. 2021-02-24]. available on the link: https://www.uptodate.com/contents/clinical-features-evaluation-and-diagnosis-of-sepsis-in-term-and-late-preterm-infants
We wanted to evaluate the difference between those two, not to suggest a new one.
Comment: Limitations of the study should be clearly stated at the end of discussion
Reply: The statements about study limitations are at the end of conclusion section (lines 334-338).

Round 2
Reviewer 2 Report
The authors performed most of the requested corrections.
However, minor aditional corrections are needed:
- Statement regarding cut-offs should be included in methodology together with reference, because the cut-offs were not calculated (also in methodology the authors should explain what was measured)
- Limitations of the study should be present at the end of discussion, not in conclusion and should be updated. Retrospective character of the study was the main limitation of the study. This should be adressed as well.
Author Response
Reply to review report
The authors thank the reviewer for valuable comments. The manuscript has been modified based on them.
Comment: Statement regarding cut-offs should be included in methodology together with reference, because the cut-offs were not calculated (also in methodology the authors should explain what was measured)
Reply: The manuscript has been modified according to the comment. Lines 260-263.
Comment: Limitations of the study should be present at the end of discussion, not in conclusion and should be updated. Retrospective character of the study was the main limitation of the study. This should be adressed as well.
Reply: The manuscript has been modified according to the comment. Lines 258-260.
